**Data Availability Statement:** The data used for this manuscript cannot be shared publicly without restrictions because of ethical and legal restrictions

# Cardiovascular vulnerability predicts hospitalisation in primary care clinically suspected and confirmed COVID-19 patients: A model development and validation study

**Florien S. van Royen**[1]*, **Linda P. T. Joosten**[1], **Maarten van Smeden**[2], **Pauline Slottje**[3], **Frans H. Rutten**[1], **Geert-Jan Geersing**[1], **Sander van Doorn**[1]

1 Department of General Practice, Julius Center for Health Sciences and Primary Care, University Medical Center Utrecht, Utrecht University, Utrecht, Netherlands, 2 Department of Epidemiology, Julius Center for Health Sciences and Primary Care, University Medical Center Utrecht, Utrecht University, Utrecht, Netherlands, 3 Department of General Practice, Academic network of general practice, Amsterdam UMC, Vrije Universiteit Amsterdam, Amsterdam, Netherlands

* f.s.vanroyen-5@umcutrecht.nl

## Abstract

### Objectives

Cardiovascular conditions were shown to be predictive of clinical deterioration in hospitalised patients with coronavirus disease 2019 (COVID-19). Whether this also holds for outpatients managed in primary care is yet unknown. The aim of this study was to determine the incremental value of cardiovascular vulnerability in predicting the risk of hospital referral in primary care COVID-19 outpatients.

### Design

Analysis of anonymised routine care data extracted from electronic medical records from three large Dutch primary care registries.

### Setting

Primary care.

### Participants

Consecutive adult patients seen in primary care for COVID-19 symptoms in the 'first wave' of COVID-19 infections (March 1 2020 to June 1 2020) and in the 'second wave' (June 1 2020 to April 15 2021) in the Netherlands.

### Outcome measures

A multivariable logistic regression model was fitted to predict hospital referral within 90 days after first COVID-19 consultation in primary care. Data from the 'first wave' was used for derivation (n = 5,475 patients). Age, sex, the interaction between age and sex, and the number of cardiovascular conditions and/or diabetes (0, 1, or ≥2) were pre-specified as candidate

applying to the human research participant data of the three included primary care databases: JGPN, ANH and AHA. Sharing of data is fostered by our research group, yet should involve a discussion within the steering groups of the separate databases, as well as consulting an ethics committee where appropriate. This is needed to ensure that data are shared in accordance with participant consent and all applicable local laws and regulations. Data are available from the Institutional Data Access (contact via SecretariaatHAG-Onderzoek@umcutrecht.nl) for researchers who meet the criteria for access to confidential data, accompanied with a protocol describing the research questions aimed to answer. URL:https://juliuscentrum.umcutrecht.nl/en/.

**Funding:** This work was supported by the Dutch Heart Foundation (grant number 2020T063). The Dutch Heart Foundation had no role in study design, data collection and analysis, decision to publish, or preparation of the manuscript.

**Competing interests:** The authors have declared that no competing interests exist.

predictors. This full model was (i) compared to a simple model including only age and sex and its interaction, and (ii) externally validated in COVID-19 patients during the 'second wave' (n = 16,693).

## Results

The full model performed better than the simple model (likelihood ratio test *p*<0.001). Older male patients with multiple cardiovascular conditions and/or diabetes had the highest predicted risk of hospital referral, reaching risks above 15–20%, whereas on average this risk was 5.1%. The temporally validated c-statistic was 0.747 (95%CI 0.729–0.764) and the model showed good calibration upon validation.

## Conclusions

For patients with COVID-19 symptoms managed in primary care, the risk of hospital referral was on average 5.1%. Older, male and cardiovascular vulnerable COVID-19 patients are more at risk for hospital referral.

## Introduction

Coronavirus disease 2019 (COVID-19), caused by SARS-CoV-2, has led to a global pandemic ever since the first cases were described in late 2019. Despite clear improvement in terms of vaccination efficacy and treatment options for hospitalised patients, COVID-19 remains a global public health burden. For instance, at present, global vaccination coverage is still low in many countries (particularly in low- and middle-income countries). Moreover, waning immunity after vaccination is already observed, notably in vulnerable individuals. In addition, new emerging virus variants–like omicron–escape protective immunity following vaccination, and finally, a substantial part (5–10% in highly vaccinated countries) of the population refuse vaccination altogether for a variety of reasons. Thus–inevitably–global circulation of SARS-SoV-2 will remain with new (seasonal) outbreaks likely to occur. COVID-19 will keep influencing the organisation of healthcare worldwide in the upcoming years, perhaps decades.

Thereby, it is pivotal to increase our knowledge on this relatively new disease, for instance in order to learn how to orchestrate the flow of patients depending on the expected course or natural prognosis of the disease. Indeed, a fast-growing amount of studies evaluated prognostic factors for prognosticating COVID-19 patients. However, most of these studies focus on an in-hospital population, with only few focusing on outpatient management [1]. This is unfortunate as the clinical presentation of (suspected) COVID-19 starts-off with initially mild to moderate symptoms in the first week of illness, and only in some with progression to hypoxemia for which hospital (or even ICU) admission is needed [2]. If such deterioration occurs, patients in the Netherlands, as in many countries worldwide, are seen in a primary care setting. The primary care physician is therefore often the first to decide on the need and optimal timing for more impactful measures, such as intensified monitoring, prescription of budesonide inhalation or perhaps novel virus inhibitors, or ultimately referral for hospitalisation [3,4]. Unfortunately, evidence based tools or knowledge to help primary care physicians on deciding *how* to triage COVID-19 patients and detect patients in need for referral from those in whom a relatively benign trajectory is to be expected are currently lacking.

From studies done in hospitalised COVID-19 patients, we know that underlying cardiovascular diseases are strong predictors for further disease deterioration towards ICU admittance or death [5–9], an association that remains relevant also in vaccinated populations [10]. If preexisting cardiovascular disease and/or diabetes also increases the risk of clinical deterioration already in primary care, this likely is instrumental to guide and orchestrate outpatient management in COVID-19. The aim of this study was therefore to determine the prognostic incremental value of cardiovascular vulnerability–defined by the number of cardiovascular diseases and/or type 2 diabetes mellitus–in predicting the risk of escalation of care (i.e. hospital referral) in primary care patients with clinically suspected or confirmed COVID-19.

## Methods

### Study design

This study involves an analysis of anonymised observational electronic medical record data of community people registered by the primary care physician with either confirmed or clinically suspected COVID-19. We assessed the incremental value of cardiovascular disease and/or diabetes by developing a prognostic prediction model in a cohort of patients from the 'first wave' of COVID-19 infections in the Netherlands (March 1 2020 to June 1 2020) that was temporally validated in a cohort of patients from the 'second wave' of infections in the Netherlands (June 1 2020 to April 15 2021). Where appropriate for this study, we adhered to the TRIPOD guideline for reporting prediction models [11].

### Databases

Patients were included from three similar ongoing and dynamic primary care databases run by the academic hospitals of the cities and surrounding municipalities of Utrecht and Amsterdam, containing pseudonymised medical data of approximately 850,000 patients in total: the Julius General Practitioner's Network (JGPN) University Medical Center Utrecht, the Academic Network of General Practice at VU University medical center in Amsterdam (ANH VUmc), and the Academic General Practitioner's Network at Academic Medical Center Amsterdam (AHA AMC) [12–14]. Two databases (JGPN and ANH VUmc) were used to identify patients for the development of the prediction model (i.e. development cohort) and all three databases (JGPN, ANH VUmc and AHA AMC) were used to identify patients for the temporal validation (i.e. the validation cohort).

### Study population and data collection

Detailed information on how data were collected for our study population is described in S1. In short, patients for the development cohort were included from March 1 2020 to June 1 2020 (the 'first wave' of the COVID-19 pandemic in the Netherlands). During this time period, very limited polymerase chain reaction (PCR) testing for COVID-19 was available, and moreover mainly restricted to more severe hospitalised cases. Consequently, many symptomatic patients consulting their primary care physician with highly suggestive of COVID-19 were not tested. We therefore included all consecutive adult patients aged 18 years or older, who visited their primary care physician with confirmed *or* suspected (based upon clinical symptoms) COVID-19.

For the validation cohort, consecutive adult patients from the 'second wave' of COVID-19 infections were included (data from June 1 2020 until April 15 2021, see S1). At this point in time, the Dutch government made PCR COVID-19 tests freely available and these were recommended for all symptomatic subjects in the Netherlands and for those who were in close

contact with a confirmed COVID-19 patient. Moreover, at that time GPs were instructed to uniformly code confirmed cases in their medical records using standardized coding. Thus, only confirmed COVID-19 cases were included in the cohort for validation of the model.

## Outcome

The primary outcome of the prediction model in this study was referral to an emergency ward for intended hospital admission. This was defined as any clinical deterioration resulting in hospital referral by the primary care physician that was recorded as such in the consultation annotation (free text) of the medical record. To capture the full spectrum of complications of COVID-19 resulting in hospitalisation, follow-up lasted 90 days after first consultation for COVID-19 suspected symptoms. To this end, all anonymised consultation texts were manually screened for any emergency hospital referral by (primary care) clinical scientists (FSvR, LPTJ, SvD, and GJG) and cases of doubt were discussed, until consensus was reached.

## Candidate predictors

Based upon existing literature from hospitalised COVID-19 patients, we a-priori specified the following candidate predictors prior to the analysis phase: age, sex, the interaction between age and sex, and the number of cardiovascular diseases. The latter was defined as (history of) type 2 diabetes mellitus, heart failure, coronary artery disease, peripheral arterial disease, stroke/ transient ischemic attack (TIA), venous thromboembolism (pulmonary embolism or deep venous thrombosis; VTE), and/or atrial fibrillation (AF). Presence of these diseases was based on the corresponding disease coding (S1 Table) at any point before the first COVID-19 consultation in the patient's medical record. The number of cardiovascular diseases were counted per patient and categorised into: no cardiovascular disease, one cardiovascular disease, or two or more cardiovascular diseases.

## Sample size

The model development cohort yielded 5,475 eligible patients with an event fraction of 0.068 (6.8%, n = 373) for the primary outcome referral to the hospital. Prior to prediction analysis, the number of allowed candidate predictors was determined. Based on the proposed calculation for sample size in prediction modelling by Riley et al. [15], the maximum number of candidate predictors that can be modelled was 30 with a $R^2$ Cox-Snell ($R^2$cs) of 0.0495. As this $R^2$cs was estimated in absence of a known value, varying $R^2$cs from 0.0395 to 0.0595 yielded a minimum of 24 and a maximum of 37 candidate predictors, including interaction terms. By using the candidate predictors age, sex, the interaction between age and sex, and the number of cardiovascular diseases with three categories, the sample size of 5,475 eligible patients was deemed sufficient and large enough for model development.

## Missing data

Candidate predictors age, sex and cardiovascular disease had no missing data. Missing values in baseline of characteristics measurements of CRP, BMI and oxygen saturation level were not imputed as these determinants were not used further in predictive modelling.

## Statistical analyses

Baseline characteristics were summarised using descriptive statistics with categorical variables as numbers with percentages and continuous variables as means with standard deviations or medians with interquartile ranges (IQR). A multivariable logistic regression modelling

approach was used to explore the predictive value of cardiovascular disease and/or diabetes–beyond age and sex–on COVID-19 prognosis. Hereto, all included patients were entered in a fixed model with the predictors i) age, ii) sex, iii) the interaction between age and sex, and iv) the categorical number of cardiovascular diseases and/or diabetes as a dummy variable (with 'no cardiovascular disease' as reference category); i.e. the full model. Next, a second model–i.e. the simple model–was fitted using only the predictors i) age, ii) sex, and iii) the interaction between age and sex. In both models, age was considered as a continuous variable and was studied using a restricted cubic spline function to account for possible non-linearity with 4 knots on the percentiles 0.05, 0.35, 0.65 and 0.95 [16]. The incremental prognostic value of the number of cardiovascular diseases and/or diabetes was assessed by comparing the full and simple model's c-statistics ($\Delta$AUC), Cox-Snell $R^2$cs ($\Delta R^2$cs), and a likelihood ratio test (alpha of 0.05 for significance). The models were internally validated using Harrell's bootstrapping with 100 repetitions to obtain optimism corrected estimates of the c-statistic, and $R^2$ and slope were calculated. For the temporal external validation, calibration and discrimination were evaluated: observed and predicted events were calculated and depicted in calibration plots and for discrimination areas under the curve (AUC/c-statistic) were calculated. Other performance measures for temporal external validation that were calculated are: calibration slope, calibration intercept, calibration in the large, $R^2$cs, and Brier score. Brier scores assess the overall goodness of fit of models, with smaller numbers indicating better performance. Confidence intervals for c-statistics were obtained using the Delong method. For $R^2$cs and Brier score confidence intervals, bootstrapping was used with repetitions set at 1000. Validation was done in the whole validation dataset as well as separately in the JGPN, ANH VUmc, and AHA AMC validation cohorts. All statistical analyses were performed in R version 4.0.3 with R base, rms, pROC, DescTools, and rmda packages [17–21].

## Ethics

This research was conducted in accordance with Dutch law and the European Union General Data Protection Regulation and according to the principles of the Declaration of Helsinki. The need for formal ethical reviewing was waived by the local medical research ethics committee of the University Medical Center Utrecht, the Netherlands as the research did not require direct patient or physician involvement. The JHN, ANH VUmc and AHA AMC databases may be used for scientific purposes and contain pseudonymised routine care data from the EMRs of all patients of the participating general practices, except those patients who objected to this. Anonymised datasets were extracted from these databases by the respective data managers for the purpose of this research.

## Results

### Patient characteristics

Patient characteristics of the (clinically suspected and confirmed COVID-19) development cohort are described in Table 1. There were 5,475 patients included in this cohort: 2,825 from JGPN and 2,650 from ANH VUmc. In ANH VUmc, 71.5% were coded as R74, 10.7% as R81, and 19.2% as R83. Differences in patient characteristics between both datasets in the development cohort were minor. Around a quarter of patients suffered from one or more cardiovascular disease, most often type 2 diabetes and coronary artery disease.

Patient characteristics of the (confirmed COVID-19) validation cohort are described in Table 2. From the total of 16,693 patients in the validation cohort 5,420 originated from JGPN, 4,989 from ANH VUmc, and 6,284 from AHA AMC. The patient characteristics in these three

**Table 1. Baseline characteristics of the development cohort of 5,475 community patients with confirmed or clinically suspected COVID-19 from the 'first wave' in the Netherlands.**

| Characteristics | JGPN (n = 2,825) | ANH VUmc (n = 2,650) |
|---|---|---|
| Age in years (IQR) | 48 (34–62) | 49 (36–62) |
| Male sex | 1144 (40.1%) | 1068 (40.3%) |
| Cardiovascular and metabolic diseases | | |
| Heart failure | 109 (3.9%) | 80 (3.0%) |
| Coronary artery disease | 230 (8.1%) | 189 (7.1%) |
| Atrial fibrillation | 139 (4.9%) | 177 (6.7%) |
| Peripheral arterial disease | 49 (1.7%) | 37 (1.4%) |
| History of stroke/TIA | 136 (4.8%) | 117 (4.4%) |
| History of VTE | 123 (4.4%) | 100 (3.8%) |
| Type 2 diabetes mellitus | 300 (10.6%) | 262 (9.9%) |
| 0 CVD/Diabetes | 2132 (75.5%) | 2053 (77.5%) |
| 1 CVD/Diabetes | 440 (15.6%) | 386 (14.6%) |
| ≥2 CVD/Diabetes | 253 (9.0%) | 211 (8.0%) |
| Other comorbidities | | |
| History of any cancer | 198 (7.0%) | 207 (7.8%) |
| Hypertension | 721 (25.5%) | 616 (23.2%) |
| Hypercholesterolemia | 261 (9.2%) | 331 (12.5%) |
| All pulmonary disease | 575 (20.4%) | 432 (16.3%) |
| COPD | 234 (8.3%) | 155 (5.8%) |
| Asthma | 434 (15.4%) | 315 (11.9%) |
| BMI in kg/m$^2$ (IQR) | 27 (24–31) (n = 1,091) | 27 (24–31) (n = 1,072) |
| Median oxygen saturation in % (IQR) | 98 (95–98) (n = 191) | 98 (96–99) (n = 87) |
| CRP in mg/L (IQR) | 6 (2–23) (n = 480) | 3 (1–9) (n = 646) |
| Hospital referrals | 185 (6.5%) | 188 (7.1%) |

BMI = Body Mass Index; CRP = C-reactive protein; CVD = cardiovascular disease; IQR = interquartile rage; TIA = transient ischemic attack; VTE = venous thromboembolism.

datasets were very similar. Around 15–20% suffered from one or more cardiovascular disease, again most often type 2 diabetes and coronary artery disease.

## Model development and internal validation

All 5,475 patients in de development cohort were used for model development. 373 patients (6.8%) had the outcome hospital referral. All predefined model regression coefficients of the full and simple models with confidence intervals are shown in Table 3. The apparent c-statistic of the full model was 0.693 (95%CI 0.665–0.721) and the internally validated c-statistic was 0.688 (95%CI 0.660–0.716). The apparent c-statistic of the simple model was 0.681 (95%CI 0.653–0.710) and the internally validated c-statistic was 0.680 (95%CI 0.652–0.708). The full and the simple model are compared in Table 4. The full model performed significantly better than the simple model (p- value for likelihood ratio test, $\chi^2$ = 19.5, $df$ = 2, $p<0.001$). Fig 1 gives a visual representation of the full model showing the predicted risks of hospital referral as a function of (increasing) age, stratified by sex and by the number of cardiovascular diseases and/or diabetes. Overall risks are higher for male patients and increase with age. Furthermore, a higher risk is observed in patients with underlying cardiovascular disease.

**Table 2. Baseline characteristics of the validation cohort of 16,693 community patients with confirmed COVID-19.**

| Characteristics | JGPN (n = 5,420) | ANH VUmc (n = 4,989) | AHA AMC (n = 6,284) |
|---|---|---|---|
| Age in years (IQR) | 43 (30–56) | 47 (34–59) | 49 (36–60) |
| Male sex | 2400 (44.3%) | 2121 (42.5%) | 2462 (39.2%) |
| Cardiovascular diseases | | | |
| Heart failure | 86 (1.6%) | 64 (1.3%) | 101 (1.6%) |
| Coronary artery disease | 262 (4.8%) | 270 (5.4%) | 346 (5.5%) |
| Atrial fibrillation | 122 (2.3%) | 139 (2.8%) | 176 (2.8%) |
| Peripheral arterial disease | 39 (0.7%) | 44 (0.9%) | 39 (0.6%) |
| History of stroke/TIA | 123 (2.3%) | 151 (3.0%) | 229 (3.6%) |
| History of VTE | 124 (2.3%) | 134 (2.7%) | 226 (3.6%) |
| Type 2 diabetes mellitus | 476 (8.8%) | 495 (9.9%) | 715 (11.4%) |
| 0 CVD/Diabetes | 4566 (84.2%) | 4059 (81.4%) | 4964 (79.0%) |
| 1 CVD/Diabetes | 586 (10.8%) | 663 (13.3%) | 955 (15.2%) |
| ≥2 CVD/Diabetes | 268 (4.9%) | 267 (5.4%) | 365 (4.8%) |
| Other comorbidities | | | |
| History of any cancer | 207 (3.8%) | 331 (6.6%) | 304 (4.8%) |
| Hypertension | 883 (16.3%) | 1,009 (20.2%) | 1,486 (23.6%) |
| Hypercholesterolemia | 309 (5.7%) | 493 (9.9%) | 527 (8.4%) |
| All pulmonary diseases | 648 (12.0%) | 610 (12.2%) | 867 (13.8%) |
| COPD | 174 (3.2%) | 186 (3.7%) | 184 (2.9%) |
| Asthma | 534 (9.9%) | 477 (9.0%) | 726 (11.6%) |
| BMI in kg/m$^2$ (IQR) | 28 (24–32) (n = 1,685) | 27 (24–32) (n = 1,823) | 29 (25–33) (n = 2,178) |
| Median oxygen saturation in % (IQR) | 98 (96–98) (n = 176) | 98 (97–99) (n = 134) | 98 (97–99) (n = 75) |
| CRP in mg/L (IQR) | 3 (2–12) (n = 545) | 2 (1–5) (n = 936) | 3 (1–10) (n = 1,055) |
| Hospital referrals | 219 (4.0%) | 187 (3.7%) | 357 (5.7%) |

BMI = Body Mass Index; CRP = C-reactive protein; CVD = cardiovascular disease; IQR = interquartile range; TIA = transient ischemic attack; VTE = venous thromboembolism.

## Temporal external validation

Predicted risks were overall slightly higher than the observed risk (6.2% versus 4.6%) and the calibration slope was 1.36. Overall discrimination showed an AUC of 0.747 (95%CI 0.729–0.764). Performance measures based on the full validation cohort and stratified by database are shown in Table 5. The overall calibration plot and the calibration plots per database separately are shown in S1 Fig. The hospital referral prevalence was lower in the validation datasets than in the development datasets (4.7% versus 6.8%).

## Discussion

Cardiovascular vulnerability is a predictor of hospital referral in a population of 5,475 consecutive adult patients in primary care with confirmed or clinically suspected COVID-19 in the 'first wave' of infections in the Netherlands. This finding was confirmed by temporal validation in a population of 16,693 consecutive confirmed COVID-19 adult primary care patients in the 'second wave', exemplifying the robustness of our inferences. On average, in the combined data from the first and second wave (n = 22,168 confirmed and clinically suspected primary care COVID-19 patients), 5.1% was referred to the hospital for considering admission. A model including the number of cardiovascular conditions and/or diabetes (0, 1, or ≥2) in addition to age and sex and the interaction between age and sex, showed moderate to good

**Table 3. Model development and internal validation using logistic regression.**

|  | Full model | | Simple model | |
|---|---|---|---|---|
| Predictor | Regression coefficient | 95% confidence interval | Regression coefficient | 95% confidence interval |
| Intercept | -4.110 | -6.338; -2.149 | -3.988 | -6.228; -2.038 |
| Age | 0.026 | -0.034; 0.092 | 0.022 | -0.037; 0.088 |
| Age' | 0.045 | -0.170; 0.251 | 0.085 | -0.127; 0.289 |
| Age" | -0.172 | -0.719; 0.389 | -0.268 | -0.809; 0.286 |
| Female sex | 0.324 | -2.358; 3.105 | 0.214 | -2.450; 2.974 |
| Interaction sex and age | -0.019 | -0.102; 0.062 | -0.016 | -0.099; 0.065 |
| Interaction sex and age' | 0.032 | -0.247; 0.314 | 0.015 | -0.262; 0.296 |
| Interaction sex and age" | -0.056 | -0.805; 0.686 | -0.014 | -0.758; 0.726 |
| CVD 1 | 0.564 | 0.280; 0.844 |  |  |
| CVD $\geq$2 | 0.636 | 0.282; 0.983 |  |  |
| *Apparent c-statistic* | 0.693 | 0.665; 0.721 | 0.681 | 0.653; 0.710 |
| *$R^2cs$* | 0.030 | 0.021; 0.039 | 0.026 | 0.018; 0.035 |
| *Internal validation c-statistic* | 0.688 | 0.660; 0.716 | 0.680 | 0.652; 0.708 |
| *Internal validation $R^2$* | 0.070 |  | 0.061 |  |
| *Internal validation slope* | 0.965 |  | 0.957 |  |

Regression coefficients with 95% confidence intervals, c-statistic, and internal validation performance measures of the full model with predictors age, sex and number of cardiovascular diseases and the simple model with only age, sex and the interaction between age and sex as predictors. Age was divided into three subgroups (shown as age, age' and age") using restricted cubic spline function to account for non-linearity. CVD = cardiovascular disease; $R^2cs$ = $R^2$ Cox-Snell.

performance and demonstrated consistent and good discrimination and calibration upon temporal external validation. The model showed a c-statistic of 0.747 (95%CI 0.729–0.764).

Although most (vaccinated) COVID-19 patients experience a favourable prognosis without the need for referral for hospital care, studies on COVID-19 are mainly focussed on those seen in the hospital setting. While on average the overall risk for hospital referral in this adult primary care cohort with COVID-19 was low (5.1%), it is much higher than the hospitalisation rate for other lower respiratory infections in primary care which is estimated at approximately 1% of the adult population affected [22]. In our study, age, sex and the number of concurrent cardiovascular conditions and/or diabetes predicted patients at far greater risk of hospital referral. In fact, for female patients without cardiovascular diseases or diabetes, the risk of hospital referral was well below 10% even in the eldest elderly (aged 80+). Contrastingly, in the presence of cardiovascular diseases and/or diabetes, patients experience higher risks already at younger ages, notably males. For instance, a male patient with two or more underlying cardiovascular diseases and/or diabetes, had a predicted risk of 15% already at the age of around 57 years and this predicted risk will even further increase to above 20% from the age of 80 onwards. This indicates the incremental effect of cardiovascular diseases and/or diabetes in

**Table 4. Comparing the full and the simple model.**

| ΔAUC | 0.012 |
|---|---|
| **ΔR$^2$cs** | 0.004 |
| **Likelihood ratio, $\chi^2$** | 19.531, df = 2, *p* = 5.740e-05 |

ΔAUC, ΔR$^2$cs and Deviance (likelihood ratio) in comparing model with and model without number of cardiovascular diseases (0, 1 or $\geq$2) modelled. ΔAUC is calculated by subtracting model 2 unadjusted c-statistic from model 1 unadjusted c-statistic. ΔR$^2$cs is calculated by subtracting model 2 R$^2$cs from model 1 R$^2$cs.

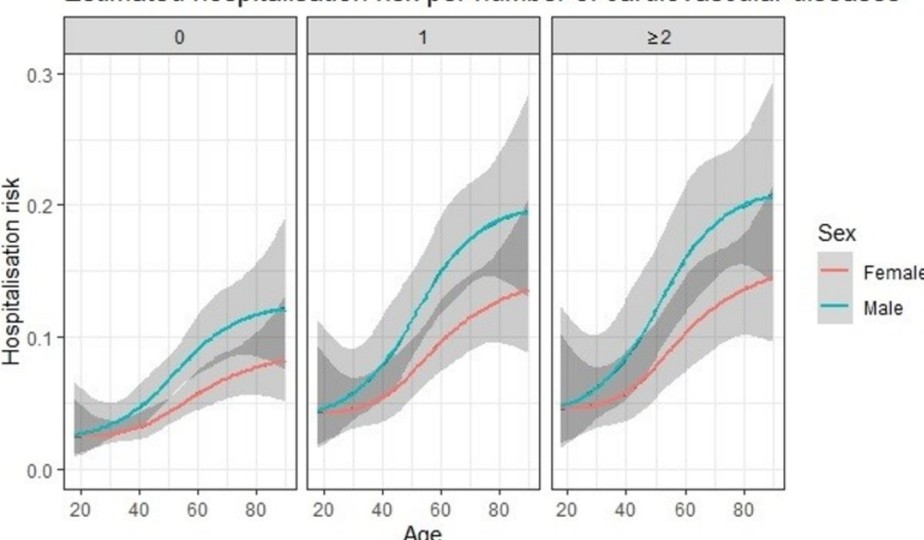

**Fig 1. Predicted risk of hospitalisation.** Plotted risk of hospital referral in (clinically suspected and confirmed) COVID-19 primary care patients at a certain age and stratified by sex and by the number of cardiovascular diseases and/or diabetes (i.e. 0, 1 or ≥2). Confidence intervals are shown in grey.

addition to age and sex in predicting the risk for complicated COVID-19 disease trajectories in primary care patients.

## Comparison with existing literature

Our findings overall confirm those from previous studies done in the hospital setting where age and male sex are important predictors for disease progression towards the endpoints ICU admission or death [1,23–25]. Social, behavioural, comorbidity and biological differences (ACE2 expression, sex-hormones, X-chromosome exposure) between male and female sexes all might contribute to the higher risks of COVID-19 progression observed in males, although probably not all mechanisms have been fully elucidated yet [26,27]. Also, in *hospitalised* patients, it has been demonstrated that there is an association between cardiovascular disease and COVID-19 complicated disease trajectories, with higher prevalence of cardiovascular

**Table 5. Performance temporal validation.**

| | Total validation population (n = 16,693) | Total patients JGPN (n = 5,420) | Total patients AHA AMC (n = 6,284) | Total patients ANH Vumc (n = 4,989) |
|---|---|---|---|---|
| Area under the curve | 0.747 (CI 0.729; 0.764) | 0.780 (CI 0.750; 0.810) | 0.718 (CI 0.691; 0.745) | 0.751 (CI 0.714; 0.789) |
| Calibration slope | 1.358 (CI 1.246; 1.471) | 1.510 (CI 1.306;1.718) | 1.218 (CI 1.050; 1.387) | 1.415 (1.190; 1.645) |
| Calibration intercept | -0.333 (CI -0.408; -0.261) | -0.396 (CI -0.535; -0.262) | -0.143 (CI -0.253; -0.036) | -0.561 (CI -0.712; -0.416) |
| Calibration in the large: Mean observed risk Mean predicted risk | 0.046 0.062 | 0.040 0.058 | 0.057 0.065 | 0.038 0.063 |
| $R^2$cs | 0.035 (CI 0.029; 0.041) | 0.040 (CI 0.030; 0.050) | 0.033 (CI 0.024; 0.042) | 0.032 (CI 0.021; 0.042) |
| Brier score | 0.042 (CI 0.039; 0.045) | 0.037 (CI 0.033; 0.041) | 0.052 (CI 0.047; 0.056) | 0.035 (CI 0.031; 0.040) |

Temporal validation performance measures with 95% confidence intervals (CI). $R^2$cs = $R^2$ Cox-Snell.

disease and diabetes described in those with critical illness [5–9,28]. Our study shows that this prognostically unfavourable effect is already present much earlier on in the COVID-19 disease course, at the start of symptoms in primary care.

This is in line with previous research, where this additive effect of (cardiovascular) comorbidities was also described by the 4C Mortality Score [29]. In this study, the authors demonstrated that the number of comorbidities, importantly including cardiovascular comorbidities, had a more predictive effect than taking only individual co-morbidities in predicting in-hospital mortality of COVID-19 patients [29]. Furthermore, there are two large community-based prediction studies also highlighting the importance of cardiovascular comorbidities as a predictor in the community COVID-19 population. The QCOVID model that was developed in the UK and recently validated in a vaccinated population, was based on data from primary care and showed a c-statistic >0.9 for the primary outcome time to death from COVID-19. The domain of that study, however, covered the whole general population *regardless* of COVID-19 diagnosis and therefore this can best be interpreted as the risk prediction of getting infected with COVID-19 *and* subsequently having complications from COVID-19. Thus, the aim of this model was to inform UK health policy and support interventions to manage COVID-19 related risks, rather than inform medical decision making during patient consultations in confirmed or clinically suspected COVID-19 cases [10,30]. With only 0.07% with the outcome death, and thus very low a priori chance, the c-statistic 'misleadingly' moves towards 1.0. Another similar public health based UK study in patients with and without COVID-19 identified determinants that were associated with COVID-19 related death in the OpenSA-FELY primary care database by linking primary care records to reported COVID-19 related deaths. It found the most predictive clinical determinants to be increasing age, male sex, type 2 diabetes mellitus, and cardiovascular disease, similar to our findings [31]. While the domain notably differs between patients seeking primary care for COVID-19 symptoms in our study and the adult community as a whole in these studies form the UK, all draw similar conclusions on the increased risk of clinical deterioration in patients with (multiple) cardiovascular disease and/or diabetes.

## Strengths and limitations

This research contributes to the evidence-based prognostication of community COVID-19. We were able to use routine primary care databases capturing both the 'first' and 'second' wave of COVID-19 infections in the Netherlands. We used state-of-the-art methodology including external temporal validation to predict clinical deterioration in a patient population that is currently understudied. The developed statistical model is not intended to be used as a clinical prediction algorithm in primary care. Conversely, the model served as a tool to explore and quantify the predictive value of cardiovascular disease and/or diabetes in the primary care COVID-19 domain. For full appreciation of our findings, however, some limitations also need to be addressed. First, the model was developed in a dataset with a low event fraction of the outcome hospital referral. Yet the number of hospital referral events did allow us to perform robust multivariable regression techniques. Second, there are limitations to using routine care registry data that could have resulted in misclassification of the study population, predictors and outcome, and most importantly it has the risk of missing values. For example, uncertainty concerning COVID-19 infection status may exist (primarily in the first wave) as COVID-19 PCR test results were not automatically linked to the primary care electronic medical records. However, the model proved its transportability in primary care patients in a different time period with satisfactory calibration and discrimination, during a time window where PCR testing was widely performed. Furthermore, the outcome hospital *referral* was based upon a

rigorous manual extraction of medical records by pairs of researchers, albeit actual hospital *admittance* was not formally confirmed based upon linkage to hospital records. Additionally, there are differences between our development and validation population: the patients from the 'first wave' are all symptomatic patients that visited their primary care physician for symptoms suggestive of COVID-19, while the patients from the 'second wave'–due to government recommendation for individuals to get tested even in the circumstance of only mild symptoms–also include more healthy people that just informed their primary care physician of their positive COVID-19 PCR status. This could also explain the lower event fraction in the validation set (4.7% versus 6.8% in the development population). Furthermore, the model still has to show its robustness in the COVID-19 vaccinated population, although it is likely that existing risk factors will still be present even if the risk of complications is lowered due to vaccination [10]. Finally, the incremental value of the number of cardiovascular diseases and diabetes on prognosticating COVID-19 was assessed in different ways; although we did observe a highly significant change in the likelihood ratio test, the delta in c-statistic and $R^2$cs was only small to modest. Possible reasons for this include the overall low risk of hospital referral in most patients in our cohort, as well as that most patients (80.2%) in fact in our cohort did not suffer from cardiovascular diseases and/or diabetes. It has been widely acknowledged that, notably in such scenario's, a change in e.g. the c-statistic is difficult to achieve.

## Clinical implications

The readily availability of the chosen primary care predictors and the clinical applicability may provide great advantages for risk profiling patient with suspected or confirmed COVID-19 in the primary care and community setting. This can have several important clinical and public health implications. First, it may be possible to identify patients that will benefit from closer monitoring and frequent follow-up at home by predicting the risk of clinical deterioration early on in the COVID-19 disease course. By intensified monitoring of higher risk patients, critical illness may be detected earlier, potentially improving prognosis. Second, risk prediction could also support advanced care planning. Informing both patients and physicians on the risk of severe illness, may help in anticipating a more stringent or more lenient management. Last, risk profiling may be used for targeting preventive measures. Additionally, experimental regiments to *treat* symptomatic COVID-19 may be addressed to high-risk patients that may benefit most. Examples include for instance treatment with budesonide, colchicine or novel virus inhibitors; such treatment options likely benefit patients most at higher prior probability of having an adverse prognosis [3,4,32]. Nevertheless, in the end, risk prediction in primary care has to prove its value in daily practice at the background of changing characteristics of this challenging COVID-19 pandemic and influences of vaccination and virus mutations. We however do hope that prognostic studies, like ours, may aid physicians and policy makers by making informed, evidence-based decisions and thereby improve patient outcomes.

## Conclusion

In this primary care population-based study, risk of clinical deterioration leading to hospital referral after suspected or confirmed COVID-19 was on average 5.1%. This risk increased with age and was higher in males compared to females. Importantly, patients with concurrent cardiovascular disease and/or diabetes had higher predicted risks and therefore, cardiovascular disease is a predictor of clinical deterioration in the primary care COVID-19 domain. Identifying those at risk for hospital referral could have clinical implications for COVID-19 early disease management in primary care.

## Supporting information

**S1 Fig. Calibration plots in individual databases.** Fig a. Calibration plot in the total validation cohort with hospitalisation as the outcome. Fig b. Calibration plot in JGPN validation cohort with hospitalisation as the outcome. Fig c. Calibration plot in AHA validation cohort with hospitalisation as the outcome. Fig d. Calibration plot in ANH validation cohort with hospitalisation as the outcome.
(DOCX)

**S1 Table. ICPC codes used in this study.** ICPC codes used for identifying the study population and individual comorbidities. COVID-19 = coronavirus disease 2019;
ICPC = International Classification of Primary Care; TIA = transient ischemic attack;
VTE = venous thromboembolism.
(DOCX)

**S1 File. Methodological details.**
(DOCX)

## Acknowledgments

The authors would like to thank the data managers from the JGPN, ANH VUmc and AHA AMC databases and all primary care physicians contributing data.

## Author Contributions

**Conceptualization:** Florien S. van Royen, Linda P. T. Joosten, Maarten van Smeden, Frans H. Rutten, Geert-Jan Geersing, Sander van Doorn.

**Data curation:** Florien S. van Royen, Linda P. T. Joosten, Pauline Slottje, Sander van Doorn.

**Formal analysis:** Florien S. van Royen, Linda P. T. Joosten, Maarten van Smeden, Sander van Doorn.

**Funding acquisition:** Frans H. Rutten, Geert-Jan Geersing.

**Investigation:** Florien S. van Royen, Linda P. T. Joosten, Frans H. Rutten, Geert-Jan Geersing, Sander van Doorn.

**Methodology:** Florien S. van Royen, Linda P. T. Joosten, Maarten van Smeden, Geert-Jan Geersing, Sander van Doorn.

**Resources:** Pauline Slottje.

**Supervision:** Maarten van Smeden, Frans H. Rutten, Geert-Jan Geersing, Sander van Doorn.

**Writing – original draft:** Florien S. van Royen.

**Writing – review & editing:** Florien S. van Royen, Linda P. T. Joosten, Maarten van Smeden, Pauline Slottje, Frans H. Rutten, Geert-Jan Geersing, Sander van Doorn.

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
