## [Decision Letter · Decision Letter 0]

19 Jan 2022

PONE-D-21-31622Cardiovascular vulnerability predicts hospitalisation in primary care clinically suspected and confirmed COVID-19 patients: a model development and validation studyPLOS ONE

Dear Dr. van Royen,

Thank you for submitting your manuscript to PLOS ONE. After careful consideration, we feel that it has merit but does not fully meet PLOS ONE’s publication criteria as it currently stands. Therefore, we invite you to submit a revised version of the manuscript that addresses the points raised during the review process.

ACADEMIC EDITOR:

The manuscript is interesting but will require further reworking and a major revision.<o:p></o:p>

While they recognize the potential interest of the subject studied, the reviewers raised a number of important issues that need to be properly addressed.

We look forward to receiving your revised manuscript.

Kind regards,

Marcelo Arruda Nakazone, M.D., Ph.D.

Academic Editor

PLOS ONE

Journal Requirements:

Reviewers' comments:

Reviewer's Responses to Questions

**Comments to the Author**

1. Is the manuscript technically sound, and do the data support the conclusions?

Reviewer #1: Yes

Reviewer #2: Yes

Reviewer #3: Yes

2. Has the statistical analysis been performed appropriately and rigorously? 

Reviewer #1: I Don't Know

Reviewer #2: Yes

Reviewer #3: I Don't Know

3. Have the authors made all data underlying the findings in their manuscript fully available?

Reviewer #1: Yes

Reviewer #2: Yes

Reviewer #3: No

4. Is the manuscript presented in an intelligible fashion and written in standard English?

Reviewer #1: Yes

Reviewer #2: Yes

Reviewer #3: Yes

5. Review Comments to the Author

Reviewer #1: the subject of the study is a hot one. It is very helpful to have a clue to who is the patient we expect hospitalization or expect a need for hospitalization or twisted course. Together with a similar study about inpatient outcome; the two studies should be regarded as predictors of COVID-19 outcome and who is the patient who needs special attention.

Reviewer #2: The authors of this study seek to build an extensive mathematical model in order to verify whether certain previous pathologies would be important in the outcome of infection by COVID-19. Thus, they compared two situations, patients from the first wave of COVID-19 and the second wave. The mathematical model was extremely well developed, but there are questions that should be commented:

- The text is extremely long and tiring for the reader. Some parts could be summarized. Other paragraphs in which the authors explain the mathematical model are not absolutely understandable.

- The comorbidities chosen in the first group of patients, such as previous cardiovascular disease, arteriopathy and diabetes, are clearly factors related to a poor prognosis in COVID-19 infection. This was absolutely clear in the statistical results. Including the higher male risk of complications and worse outcome. Finally, the authors present a very interesting work, but they need to decide if the objective of this study is to demonstrate it is a successful mathematical model, or they want to demonstrate the factors of a worse outcome. The latter have already been described in many other previous clinical works.

In this way, I believe that the study should be written again in a clear, objective way and that what is intended to be demonstrated in this study is clearly defined.

Reviewer #3: This was a well written and thought provoking article. Within the limitations the authors should clarify that while the factors included were associated with hospital referral, this is an observational study and potential confounders should be considered.

As a clinician, I am unable to comment on the accuracy of the statistical analyses performed.

6. PLOS authors have the option to publish the peer review history of their article (what does this mean?). If published, this will include your full peer review and any attached files.

Reviewer #1: **Yes: **Aly Ahmed Abdel Rahim

Reviewer #2: No

Reviewer #3: No

---

## [Author Response · Author response to Decision Letter 0]

3 Feb 2022

Reviewer #1: the subject of the study is a hot one. It is very helpful to have a clue to who is the patient we expect hospitalization or expect a need for hospitalization or twisted course. Together with a similar study about inpatient outcome; the two studies should be regarded as predictors of COVID-19 outcome and who is the patient who needs special attention.

Answer: We thank the reviewer for these positive remarks. Indeed, with the results of our study clinicians can better predict and earlier detect community patients with COVID-19 who have a higher probability of experiencing clinical deterioration or a ‘twisted course’. We believe the comment from reviewer 1 does not raise additional questions or issues that need clarification.

Reviewer #2: The authors of this study seek to build an extensive mathematical model in order to verify whether certain previous pathologies would be important in the outcome of infection by COVID-19. Thus, they compared two situations, patients from the first wave of COVID-19 and the second wave. The mathematical model was extremely well developed, but there are questions that should be commented:

- The text is extremely long and tiring for the reader. Some parts could be summarized. Other paragraphs in which the authors explain the mathematical model are not absolutely understandable.

Answer: First, we want to thank the reviewer for the positive remarks on our model. In answer to the question; We can appreciate that indeed some parts of our manuscript may be perceived as a long read. This was done for the purpose of full transparency of the advanced methods used for model development and the validation approach. But we agree, the full description of the selection of patients and predictors, the details of the three different databases and the two different time points (‘first’ and ‘second’ wave of COVID-19 infections) may be considered complex or long for some readers. Therefore, prompted by the remark of this reviewer, and to increase readability for clinical readers, we have moved some of the details from the methods section to the appendix. Furthermore, we have made the discussion more concise, and clarified the wording in the methods section. 

All these changes are marked using track changes in the manuscript.

- The comorbidities chosen in the first group of patients, such as previous cardiovascular disease, arteriopathy and diabetes, are clearly factors related to a poor prognosis in COVID-19 infection. This was absolutely clear in the statistical results. Including the higher male risk of complications and worse outcome. Finally, the authors present a very interesting work, but they need to decide if the objective of this study is to demonstrate it is a successful mathematical model, or they want to demonstrate the factors of a worse outcome. The latter have already been described in many other previous clinical works.

In this way, I believe that the study should be written again in a clear, objective way and that what is intended to be demonstrated in this study is clearly defined.

Answer: We agree with the reviewer that our aim could be defined (even) more clearly in the manuscript. The model that we developed is not intended to be used as a clinical prediction algorithm. Contrarily, the mathematical model served only as a tool to explore and quantify the incremental predictive value on top of age and sex of concurrent cardiovascular disease on prognosis of community people with COVID-19. For that purpose, we used state-of-the-art methodological techniques including external validation, in order to be able to draw valid conclusions on the predictive value of cardiovascular disease. Thereby, our model can be used to illustrate the probability of experiencing an unfavorable outcome in community people with COVID-19, whereby our model estimates these probabilities conditional on the presence (or absence) of (the number of) cardiovascular comorbidities, age and sex. 

We also agree that indeed for secondary care COVID-19 patients the predictive value of cardiovascular disease has already been studied in numerous studies. However, even after almost two years of COVID-19 research, such a predictive effect was not yet formally evaluated in community people with COVID-19. The exploration of this incremental value of cardiovascular comorbidity on COVID-19 prognosis in this domain obviously is needed as well because the large majority (95% or more) of patients with COVID-19 will not need hospitalisation. Given the large differences in case mix between primary and secondary care a study as ours was obligatory. Moreover, the early identification of high-risk patients in primary care has paramount practical and prognostic impact because risks may be mitigated in these patients e.g. by close monitoring or the prescription of e.g. novel virus inhibitors. 

To present our aims more clearly in the manuscript, we have made a few adjustments to our formulation of the research question in the introduction and to our conclusion. Furthermore, we have added the intended use of the prediction model (as a mathematical tool only) to our strengths and limitation section of the discussion. All changes are marked using track changes. 

Reviewer #3: This was a well written and thought-provoking article. Within the limitations the authors should clarify that while the factors included were associated with hospital referral, this is an observational study and potential confounders should be considered.

As a clinician, I am unable to comment on the accuracy of the statistical analyses performed.

Answer: Indeed, cardiovascular disease and/or diabetes are not the only predictors of clinical deterioration in COVID-19 patients. From studies mainly performed in the secondary care COVID-19 domain we know that sex and age are the strongest predictors in predicting adverse COVID-19 outcomes, and that additionally also e.g. the use of immunosuppressant medication could be considered, amongst other predictors described in the literature. (Ref: https://www.bmj.com/content/369/bmj.m1328)

However, cardiovascular disease has been shown to be an important predictor of clinical deterioration in secondary care COVID-19 patients. Our aim was to explore whether this also holds in community people. The model that we developed was used to firmly illustrate (and confirm) this predictive value of cardiovascular disease on top of age and sex in primary care COVID-19 patients. We therefore only included age, sex and cardiovascular disease as predictors. Adjustments have been made to further clarify our aim as has been described in the answer to the second question by reviewer 2.

Finally, confounding will not be an issue in our work as our aim was only prognostic: to quantify the prognostic impact of these concurrent comorbidities on the prognosis of COVID-19 in primary care. Our aim was not to explore a causal relationship between cardiovascular disease and COVID-19 complications. This was done deliberately as we believe quantifying this prognostic impact on – as was done in our study – the probability of hospital referral in patients with COVID-19 seems clinically more relevant than an exploration of underlying causal mechanisms or pathways.

---

## [Decision Letter · Decision Letter 1]

28 Mar 2022

Cardiovascular vulnerability predicts hospitalisation in primary care clinically suspected and confirmed COVID-19 patients: a model development and validation study

PONE-D-21-31622R1

Dear Dr. van Royen,

We’re pleased to inform you that your manuscript has been judged scientifically suitable for publication and will be formally accepted for publication once it meets all outstanding technical requirements.

Kind regards,

Marcelo Arruda Nakazone, M.D., Ph.D.

Academic Editor

PLOS ONE

Additional Editor Comments (optional):

Reviewers' comments:

Reviewer's Responses to Questions

**Comments to the Author**

1. If the authors have adequately addressed your comments raised in a previous round of review and you feel that this manuscript is now acceptable for publication, you may indicate that here to bypass the “Comments to the Author” section, enter your conflict of interest statement in the “Confidential to Editor” section, and submit your "Accept" recommendation.

Reviewer #1: All comments have been addressed

Reviewer #2: All comments have been addressed

2. Is the manuscript technically sound, and do the data support the conclusions?

Reviewer #1: Yes

Reviewer #2: Yes

3. Has the statistical analysis been performed appropriately and rigorously? 

Reviewer #1: I Don't Know

Reviewer #2: Yes

4. Have the authors made all data underlying the findings in their manuscript fully available?

Reviewer #1: Yes

Reviewer #2: Yes

5. Is the manuscript presented in an intelligible fashion and written in standard English?

Reviewer #1: Yes

Reviewer #2: Yes

6. Review Comments to the Author

Reviewer #1: (No Response)

Reviewer #2: The authors responded adequately to the suggestions made. Now the text is understandable and absolutely clear, including understanding what exactly the authors accomplished. Again, I commend the methodology used. In short, there was a major transformation from a confusing text into an excellent study.

7. PLOS authors have the option to publish the peer review history of their article (what does this mean?). If published, this will include your full peer review and any attached files.

Reviewer #1: **Yes: **Aly Ahmed Abdel Rahim

Reviewer #2: No

---

## [Editor Report · Acceptance letter]

31 Mar 2022

PONE-D-21-31622R1 

Cardiovascular vulnerability predicts hospitalisation in primary care clinically suspected and confirmed COVID-19 patients: A model development and validation study 

Dear Dr. van Royen:

I'm pleased to inform you that your manuscript has been deemed suitable for publication in PLOS ONE. Congratulations! Your manuscript is now with our production department. 

Kind regards, 

on behalf of

Professor Marcelo Arruda Nakazone 

Academic Editor

PLOS ONE